# EQ-VAE: Equivariance Regularized Latent Space for Improved Generative Image Modeling

Theodoros Kouzelis [1 2]   Ioannis Kakogeorgiou [1]   Spyros Gidaris [3]   Nikos Komodakis [1 4 5]

## Abstract

Latent generative models have emerged as a leading approach for high-quality image synthesis. These models rely on an autoencoder to compress images into a latent space, followed by a generative model to learn the latent distribution. We identify that existing autoencoders lack equivariance to semantic-preserving transformations like scaling and rotation, resulting in complex latent spaces that hinder generative performance. To address this, we propose EQ-VAE, a simple regularization approach that enforces equivariance in the latent space, reducing its complexity without degrading reconstruction quality. By fine-tuning pre-trained autoencoders with EQ-VAE, we enhance the performance of several state-of-the-art generative models, including DiT, SiT, REPA and MaskGIT, achieving a $\times 7$ speedup on DiT-XL/2 with only five epochs of SD-VAE fine-tuning. EQ-VAE is compatible with both continuous and discrete autoencoders, thus offering a versatile enhancement for a wide range of latent generative models. Project page and code: https://eq-vae.github.io/.

## 1. Introduction

Latent generative models (Rombach et al., 2022) have become a dominant framework for high-fidelity image synthesis, achieving state-of-the-art results across diffusion models (Rombach et al., 2022; Yao et al., 2024; Ma et al., 2024), masked generative modeling (Chang et al., 2022; Li et al., 2023), and autoregressive models (Esser et al., 2021; Li et al., 2024; Tian et al., 2024). These models operate in two phases. First, an autoencoder compresses high-dimensional images into a lower-dimensional latent space, which can be continuous (e.g., SD-VAE for diffusion (Rombach et al., 2022)) or discrete (e.g., VQ-GAN for autoregressive (Esser et al., 2021; Yu et al., 2022b) and masked generative modeling (Chang et al., 2022)). This latent space retains essential semantic and structural information while discarding high-frequency details. Second, a generative model learns to model the distribution of these latent representations, enabling the synthesis of visually coherent images. At inference time, the generative model first samples a latent code, which is then decoded back into the image space by the autoencoder. While much research has focused on improving the generative phase—through advances in architectures (Peebles & Xie, 2023), objectives (Ma et al., 2024), and optimization techniques (Yao et al., 2024)—the autoencoder's role in shaping the latent space remains equally critical to overall performance.

In fact, the quality of the latent space is pivotal, influencing both computational efficiency (by reducing dimensionality and accelerating convergence in the generative phase) and the model's ability to produce high-fidelity outputs (Rombach et al., 2022). In diffusion models, most state-of-the-art approaches—such as DiT (Peebles & Xie, 2023), SiT (Ma et al., 2024), PixArt (Chen et al., 2024), SD3 (Esser et al., 2024), and Flux (Black Forest Labs, 2023)—rely on autoencoders with architectures and training objectives similar to the SD-VAE introduced in Latent Diffusion Models (LDM) (Rombach et al., 2022). LDM explores two widely adopted regularization strategies: a continuous variational approach and a discrete codebook framework. The variational approach uses a KL divergence term to align the latent distribution with a Gaussian prior, promoting a smooth and structured latent space (Kingma & Welling, 2014). Alternatively, the discrete codebook framework constrains the latent space to a finite set of learned embeddings, limiting its complexity and providing a different form of regularization (Esser et al., 2021).

These regularization strategies inherently introduce a trade-off. Stronger regularization, such as increasing the weight of the KL divergence term, produces a smoother and more learnable latent space for the generative model in the second phase (Tschannen et al., 2025). However, it also reduces the

---

[1]Archimedes,Athena Reaserch Center, Greece [2]National Technical University of Athens, Greece [3]valeo.ai, France [4]University of Crete, Greece [5]IACM-Forth, Greece. Correspondence to: Theodoros Kouzelis <theodoros.kouzelis@athenarc.gr>.

*Proceedings of the $42^{nd}$ International Conference on Machine Learning*, Vancouver, Canada. PMLR 267, 2025. Copyright 2025 by the author(s).

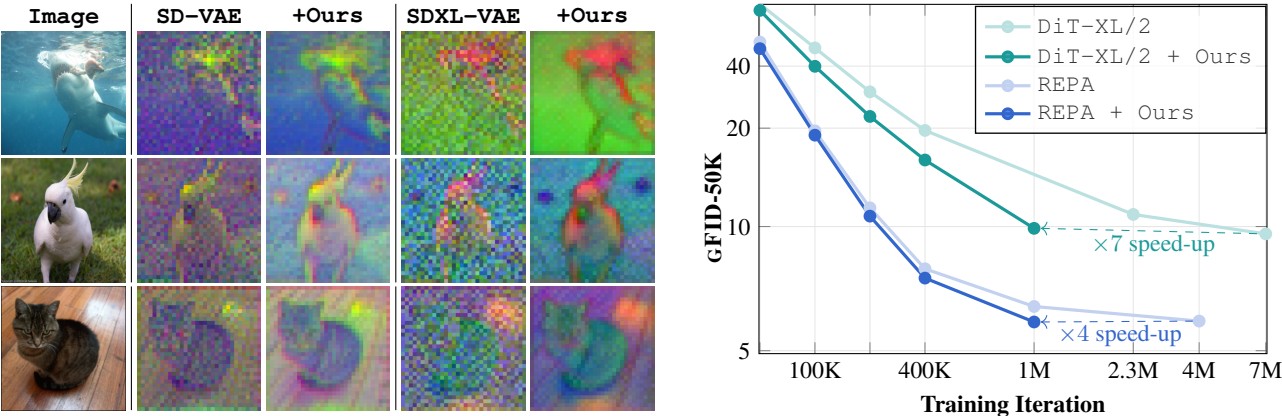

Figure 1: **Latent Space Structure (Left)** Top three principal components of `SD-VAE` and `SDXL-VAE`, with and without `EQ-VAE`, demonstrating visually that our regularization produces smoother latent representations without compromising reconstruction (See Table 1). **Accelerated Training (Right)** Training curves (without classifier-free guidance) for `DiT-XL/2` and `REPA (w/ SiT-XL/2)`, showing that our `EQ-VAE` accelerates convergence by ×7 and ×4, respectively.

information capacity of the latent representation, leading to a loss of fine-grained details and ultimately degrading reconstruction quality. Empirical evidence suggests that this trade-off can set an upper bound on the overall performance of latent generative models (Rombach et al., 2022), as the autoencoder's limited capacity to preserve detailed information restricts the overall ability of latent generative models to synthesize highly-fidelity images. This raises a fundamental question: *Can we mitigate this trade-off, creating a latent space that is more optimized for generative modeling, without compromising reconstruction quality, thereby improving the overall generative modeling process?*

A key aspect that could address this challenge lies in the structure and properties of the latent space itself. In particular, we identify an essential limitation of current state-of-the-art autoencoders: their latent representations are not equivariant to basic spatial transformations, such as scaling and rotation (see Figure 2; extended discussion in Sec. 3.2). This introduces unnecessary complexity into the latent manifold, forcing the generative model to learn nonlinear relationships that could otherwise be avoided.

To address this issue, we propose a simple yet effective modification to the training objective of autoencoders that encourages latent spaces to exhibit the aforementioned equivariance. Our method called `EQ-VAE`, penalizes discrepancies between reconstructions of transformed latent representations and the corresponding transformations of input images. Notably, `EQ-VAE` requires no architectural changes to existing autoencoder models and does not necessitate training from scratch. Instead, fine-tuning pre-trained autoencoders for a few epochs with `EQ-VAE` suffices to imbue the latent space with equivariance properties, reducing its complexity (see Figure 1-left; quantitative results in Table 5) and

facilitating learning for generative models (e.g., Figure 1-right). This is achieved without degrading the autoencoder's reconstruction quality.

Our method is compatible with both continuous and discrete autoencoders, enabling broad applicability across latent generative models. For example, applying `EQ-VAE` to the continuous `SD-VAE` (Rombach et al., 2022) significantly improves the performance of downstream diffusion models such as `DiT` (Peebles & Xie, 2023), `SiT` (Ma et al., 2024), and `REPA` (Yu et al., 2025), as measured by FID scores. Similarly, applying `EQ-VAE` to discrete `VQ-GAN` (Esser et al., 2021) enhances performance in the masked generative modeling approach `MaskGIT` (Chang et al., 2022).

We make the following contributions:

- We identify that the latent space of established autoencoders lacks equivariance under spatial transformations, which impedes latent generative modeling. Building on this observation, we propose `EQ-VAE`, a simple regularization strategy that improves generative performance without compromising reconstruction quality.
- Our method is compatible with both continuous and discrete autoencoders, enabling a plug-and-play approach for commonly used generative models such as diffusion and masked generative models.
- We show that by fine-tuning well-established autoencoders with our objective, we significantly accelerate the training of latent generative models. For instance, fine-tuning `SD-VAE` for just 5 epochs yields a ×7 speedup on `DiT-XL/2` and ×4 speedup on `REPA (w/ SiT-XL/2)` (see Figure 1 (right)).

## 2. Related work

**Autoencoders for Latent Generative Models**  Training diffusion models directly in pixel space is computationally inefficient, as most of the bits in a digital image correspond to subtle details with little perceptual significance. To overcome this issue, Rombach et al. (2022) propose latent diffusion models that operate in a compressed latent space produced in a separate stage by an autoencoder. Their KL-regularized autoencoder, SD-VAE, has been extensively utilized in numerous diffusion models (Yao et al., 2024; Ma et al., 2024; Chen et al., 2024). Subsequent research has primarily focused on minimizing the reconstruction error that sets an upper bound on generative performance, by increasing the number of latent channels (Esser et al., 2024; Black Forest Labs, 2023; Dai et al., 2023) and incorporating task specific priors (Zhu et al., 2023). To enable efficient training on high-resolution images Xie et al. (2025) and Chen et al. (2025) extensively increase the compression ratio without compromising the reconstruction quality. Hu et al. (2023) investigate the ideal latent space for generative models and find that a relatively weak decoder produces a latent distribution that enhances generative performance. Discrete autoencoders are initially introduced with VQ-VAE (van den Oord et al., 2017) to quantize image patches into discrete visual tokens. VQ-GAN (Esser et al., 2021) further refines VQ-VAE by integrating adversarial and perceptual losses, enabling more accurate and detailed representations. Subsequent works have focused on architectural improvements (Yu et al., 2022a), strategies to increase the codebook size and maximize its utilization (Yu et al., 2024; Zhu et al., 2024a). Unlike these prior approaches, we investigate a novel perspective—leveraging spatial equivariance—to shape a latent space better suited for generative modeling.

**Auxiliary Objectives and Regularization in VAEs**  Autoencoders are designed to learn latent spaces that compactly represent meaningful features of the observed data. However, without any regularization, their latent code lacks meaningful structure. Variational Autoencoders (VAEs) were introduced in Kingma & Welling (2014) to address this by minimizing the KL divergence between the latent distribution and a Gaussian prior. Many subsequent works have adopted and extended this framework (Higgins et al., 2016; Dilokthanakul et al., 2016; Tomczak & Welling, 2018; Takahashi et al., 2019). Other works have proposed alternative regularizations based on the Wasserstein distance (Tolstikhin et al., 2018; Kolouri et al., 2018), adversarial objectives (Zhao et al., 2018; Makhzani et al., 2015) and vector quantization (VQ) (van den Oord et al., 2017). Closely related to our work, Sinha & Dieng (2021) proposes a consistency regularization enforcing the latent code to be invariant under spatial transformations. Our EQ-VAE promotes *equivariance* rather than invariance under spatial transformations

and we extensively demonstrate the impact of equivariance regularization on latent generative modeling.

**Equivariance in Computer Vision**  The success of Convolutional neural networks (CNN) in numerous computer vision tasks can be largely attributed to their approximate translation equivariance that arises due to the nature of convolution. To incorporate other symmetries in the data, various group-equivariant convolutional networks have been proposed, including roto-translation equivariance in 2D (Cohen & Welling, 2016; Marcos et al., 2017; Hoogeboom et al., 2018; Weiler & Cesa, 2019), extensions in 3D (Worrall & Brostow, 2018; Thomas et al., 2018; Kondor, 2018), and scale equivariance (Rahman & Yeh, 2023; Sosnovik et al., 2020). The derivation of group equivariance constraint typically results in steerable filters constructed from a basis. Besides architectural constraints, equivariance can be achieved by parameter sharing (Ravanbakhsh et al., 2017), frame averaging (Puny et al., 2022), and canonicalization functions (Kaba et al., 2023). For autoencoder models, Winter et al. (2022) produce latent representations that are separated into a group invariant and equivariant part. Closely related to our work Ryu (2024), train autoencoders to be equivariant under horizontal and vertical flips. However, they do not investigate the impact of equivariant representations on latent generative modeling.

## 3. Method

This section presents our methodology. We first provide an overview of autoencoder models for latent generative modeling (Sec. 3.1), focusing on the continuous case used in diffusion models. We then highlight the lack of equivariance in latent representations (Sec. 3.2) and introduce EQ-VAE to address it (Sec. 3.3).

### 3.1. Preliminary: Continuous Autoencoders for Latent Generative Modeling

The first modeling stage consists of an autoencoder that compresses the pixel space into a continuous (Rombach et al. (2022)) or discrete (Esser et al. (2021)) latent space. We focus here on the continuous case. Given an input image $\mathbf{x} \in \mathbb{R}^{H \times W \times 3}$, an encoder $\mathcal{E}$ transforms the image into a compressed representation $\mathbf{z} = \mathcal{E}(\mathbf{x}) \in \mathbb{R}^{\frac{H}{f} \times \frac{W}{f} \times c}$, where $f$ is the compression ratio and $c$ are the latent channels. Then a decoder $\mathcal{D}$ takes as input the latent representation and reconstructs the image $\hat{\mathbf{x}} = \mathcal{D}(\mathbf{z})$. For an input image $\mathbf{x}$ the training objective reads as follows:

$$\mathcal{L}_{\text{VAE}}(\mathbf{x}) = \mathcal{L}_{rec}(\mathbf{x}, \hat{\mathbf{x}}) + \lambda_{gan}\mathcal{L}_{gan}(\hat{\mathbf{x}}) + \lambda_{reg}\mathcal{L}_{reg} \quad (1)$$

where $\mathcal{L}_{rec}$ consists of a pixel space reconstruction objective and a perceptual loss such LPIPS (Zhang et al., 2018), $\mathcal{L}_{gan}$ is a patch-based adversarial loss (Isola et al., 2017) and $\mathcal{L}_{reg}$

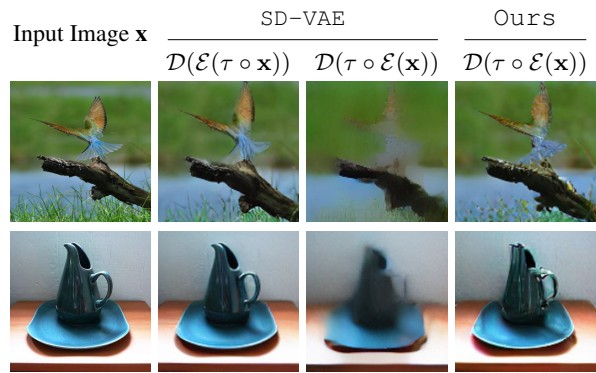

Figure 2: **Latent Space Equivariance.** Reconstructed images using `SD-VAE` (Rombach et al., 2022) and our `EQ-VAE` when applying scaling transformation $\tau$, with factor $s = 0.5$, to the input images $\mathcal{D}(\mathcal{E}(\tau \circ \mathbf{x}))$ versus directly to the latent representations $\mathcal{D}(\tau \circ \mathcal{E}(\mathbf{x}))$. Our approach preserves reconstruction quality under latent transformations, whereas `SD-VAE` exhibits significant degradation. See Figure 6 for additional examples.

is usually a Kullback-Leibler regularization with a Gaussian prior (Kingma & Welling, 2014).

### 3.2. Lack of Equivarance under Spatial Tansformations

Our work is motivated by a key observation: state-of-the-art autoencoders, such as `SD-VAE` (Rombach et al., 2022), produce latent representations $\mathcal{E}(\mathbf{x})$ that are not equivariant under basic spatial transformations like scaling and rotation.

We formalize this as follows:

**Spatial Transformation** Let $\mathbf{x}(\mathbf{p}) : \mathbb{R}^2 \to \mathbb{R}^c$ be an image (or latent representation) defined over 2D coordinates $\mathbf{p} = [u, v]^\top$. A spatial transformation $\tau \in \mathbb{R}^{2 \times 2}$ acts on the coordinates $p$ transforming $\mathbf{x}$ as follows:

$$\mathbf{x}_\tau(\mathbf{p}) = \mathbf{x}(\tau^{-1}\mathbf{p}), \qquad (2)$$

denoted compactly for all $\mathbf{p}$ as $\tau \circ \mathbf{x}$.

**Equivariance** A latent representation $\mathcal{E}(\mathbf{x})$ is equivariant with a transformation $\tau$ of the input image $\mathbf{x}$ if the transformation can be transferred to the representation output:

$$\forall \mathbf{x} \in \mathcal{X} : \quad \mathcal{E}(\tau \circ \mathbf{x}) = \tau \circ \mathcal{E}(\mathbf{x}). \qquad (3)$$

To test whether the latent representations of autoencoder models are equivariant under spatial transformations, we applied scaling and rotations $\tau$ directly to the latent code and evaluated the corresponding reconstructions. Specifically, we compare decoding transformed latent representations, $\mathcal{D}(\tau \circ \mathcal{E}(\mathbf{x}))$, to decoding latents of transformed input images, $\mathcal{D}(\mathcal{E}(\tau \circ \mathbf{x}))$. We present qualitative and quantitative results in Figure 2 and Figure 3 respectively.

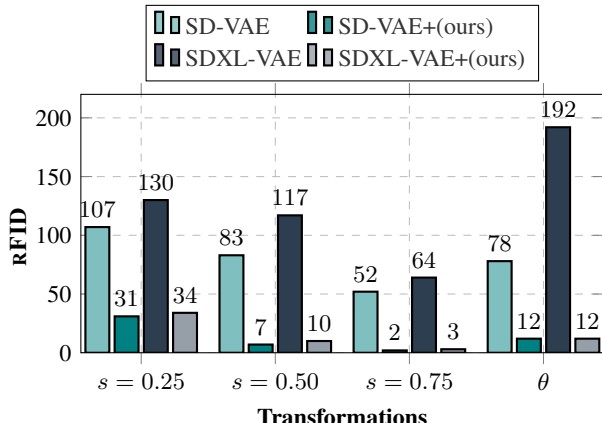

Figure 3: **Enhanced Reconstruction under Latent Transformations.** Reconstruction RFID measured between $\tau \circ \mathbf{x}$ and $\mathcal{D}(\tau \circ \mathcal{E}(\mathbf{x}))$ for various spatial transformations. We consider scaling transforms with factors $s = 0.75, 0.50, 0.25$ and also measure the average RFID over rotation angles $\theta = \frac{\pi}{2}, \pi, \frac{3\pi}{2}$. Results for `SD-VAE` (Rombach et al., 2022) and `SDXL-VAE` (Podell et al., 2024), with and without `EQ-VAE`. Our approach significantly reduces RFID compared to baselines, improving image fidelity under latent transformations. For readability, we show $\lfloor \text{RFID} \rfloor$.

Our findings reveal a clear disparity: while autoencoders reconstruct images accurately when transformations are applied to the input (i.e., $\mathcal{D}(\mathcal{E}(\tau \circ \mathbf{x}))$), applying transformations directly to the latent representation (i.e., $\mathcal{D}(\tau \circ \mathcal{E}(\mathbf{x}))$) leads to significant degradation in reconstruction quality.

This limitation arises because (1) convolutional architectures commonly used in the autoencoders of latent generative models, such as `SD-VAE`, are not equivariant under arbitrary spatial transformations such as scaling and rotation, and (2) their standard training objectives (for example, reconstruction loss and KL divergence) do not explicitly or implicitly encourage equivariance. As a result, semantically similar inputs, such as an image $\mathbf{x}$ and its scaled counterpart $\tau \circ \mathbf{x}$, are encoded into latent codes $\mathcal{E}(\mathbf{x})$ and $\mathcal{E}(\tau \circ \mathbf{x})$ that are not related by the corresponding spatial transformation, i.e. $\mathcal{E}(\tau \circ \mathbf{x}) \neq \tau \circ \mathcal{E}(\mathbf{x})$, thus unnecessarily complicating the structure of the latent space.

### 3.3. EQ-VAE: Regularization via equivariance constraints

To address this limitation, we propose `EQ-VAE`, which regularizes the latent representations to promote equivariance under spatial transformations. As seen in Figure 1 (left) this produces smoother latent representations, enabling more efficient learning.

**Explicit Regularization.** A direct way to enforce equivari-

ance is to include the equivariance constraint from Equation (3) as a loss term during training:

$$\mathcal{L}_{\text{explicit}}(\mathbf{x}) = \|\tau \circ \mathcal{E}(\mathbf{x}) - \mathcal{E}(\tau \circ \mathbf{x})\|_2^2, \qquad (4)$$

where $\tau$ is sampled from a set of spatial transformations. However, minimizing this loss alone can lead to trivial solutions, such as collapsing the latent representation to a constant value $\mathcal{E}(\mathbf{x}) = c, \ \forall \mathbf{x}$, which we observe in our experiments (see Table 7), making explicit regularization ineffective.

**Implicit Regularization.** To overcome this limitation of explicit regularization, we adopt an implicit approach. Inspired by the findings in Figure 2, this approach aligns the reconstructions of transformed latent representations $(\mathcal{D}(\tau \circ \mathcal{E}(\mathbf{x})))$ with the corresponding transformed inputs $(\tau \circ \mathbf{x})$. Specifically, we modify the original training objective of Equation (1) as follows:

$$\mathcal{L}_{\text{EQ-VAE}}(\mathbf{x}, \tau) = \mathcal{L}_{rec}\Big(\tau \circ \mathbf{x}, \mathcal{D}\big(\tau \circ \mathcal{E}(\mathbf{x})\big)\Big) + \qquad (5)$$

$$\lambda_{gan}\mathcal{L}_{gan}\Big(\mathcal{D}\big(\tau \circ \mathcal{E}(\mathbf{x})\big)\Big) + \lambda_{reg}\mathcal{L}_{reg}$$

where the changes compared to Eq. (1) are highlighted in color. Notice that when $\tau$ is the identity transformation, this formulation reduces to the original objective in Eq. (1). By leveraging the rich supervision signal from both reconstruction and adversarial objectives, this approach implicitly encourages the encoder to produce equivariant latent representations while avoiding mode collapse (see Sec. A.1).

**Transformation Design.** We focus on two types of spatial transformations: anisotropic scaling and rotations. These are parameterized as:

$$\mathbf{S}(s_x, s_y) = \begin{bmatrix} s_x & 0 \\ 0 & s_y \end{bmatrix}, \quad \mathbf{R}(\theta) = \begin{bmatrix} \cos\theta & -\sin\theta \\ \sin\theta & \cos\theta \end{bmatrix} \quad (6)$$

The final transformation is the composition of scaling and rotation: $\tau = \mathbf{S}(s_x, s_y) \cdot \mathbf{R}(\theta)$. We sample uniformly $0.25 < s_x, s_y < 1$, and $\theta \in (\frac{\pi}{2}, \pi, \frac{3\pi}{2})$. We consider these three rotation angles (multiples of $90°$) to avoid corner artifacts. For downsampling, we use bicubic interpolation. Empirically, we find *scaling equivariance is more beneficial* for generation than rotation equivariance (see Table 5).

To preserve the prior reconstruction capabilities of the autoencoder, we return to the standard objective (Eq. (1)) by sampling the identity transform $\tau = \mathbf{I}$ in Eq. (5) with probability $p_\alpha$. Our total objective can thus be written as:

$$\mathcal{L}_{total}(\mathbf{x}) = \begin{cases} \mathcal{L}_{\text{VAE}}(\mathbf{x}) & p < p_\alpha, \\ \mathcal{L}_{\text{EQ-VAE}}(\mathbf{x}, \tau) & p \geq p_\alpha. \end{cases} \quad (7)$$

where $p$ is sampled uniformly from $[0, 1]$. This controls the strength of our regularization. By default we set $p_\alpha = 0.5$ (we ablate regularization strength in Sec. A.2).

We note that our approach enforces equivariance by applying transformations directly to the latent space, distinguishing it from methods relying on input data augmentation (Brehmer et al., 2024).

**Extending EQ-VAE to Discrete Autoencoders.** So far, we described EQ-VAE in the context of continuous autoencoders. In discrete autoencoders e.g., VQ-GAN (Esser et al., 2021), the encoder outputs continuous features $\mathcal{E}(\mathbf{x})$ that are mapped to the nearest entry in a learned codebook, forming a discretized latent space via quantization. Adapting our method for discrete autoencoders, such as VQ-GAN, is straightforward. We employ our equivariance regularization loss as described in Sec. 3.3 and apply the transformations $\tau$ on the latent features $\mathcal{E}(\mathbf{x})$ *before* the quantization.

## 4. Experiments

### 4.1. Setup

**Implementation Details** We finetune all autoencoders on OpenImages to adhere to the framework used in LDM (Rombach et al., 2022). We finetune for 5 epochs with batch size 10. Detailed specifications of each autoencoder, including spatial compression rates and latent channels, are provided in Appendix E. For DiT (Peebles & Xie, 2023), SiT (Ma et al., 2024) and REPA (Yu et al., 2025), we follow their default settings and train on ImageNet (Deng et al., 2009) with a batch size of 256, where each image is resized to $256 \times 256$. We use B/2, XL/2 architectures which employ a patch size 2, except for the experiment with SD-VAE-16 in Table 1 in which we used B/1, due to its lower spatial resolution compared to other autoencoders. These models are originally trained in the latent distribution of SD-VAE-FT-EMA[1] a subsequent version of the original SD-VAE that has been further fine-tuned with an exponential moving average on LAION-Aesthetics (Schuhmann et al., 2022) (see Table 6 and (Peebles & Xie, 2023) for their performance differences). For MaskGIT, we follow (Besnier & Chen, 2023) and train on ImageNet for 300 epochs with a batch size of 256. We follow ADM (Dhariwal & Nichol, 2021) for all data pre-processing protocols.

**Evaluation** For generative performance, we train latent generative models on the latent distribution of each autoencoder and we report Frechet Inception Distance (FID) (Heusel et al., 2017), sFID (Nash et al., 2021), Inception Score (IS) (Salimans et al., 2016), Precision (Pre.) and Recall (Rec.) (Kynkäänniemi et al., 2019) using $50,000$ samples and following ADM evaluation protocol (Dhariwal & Nichol, 2021). To evaluate reconstruction, we report FID, Peak Signal-to-Noise Ratio (PSNR), Structural Similarity (SSIM) (Wang et al., 2004), and Perceptual Similarity (LPIPS) (Zhang et al., 2018) using the ImageNet validation

---

[1]https://huggingface.co/stabilityai/sd-vae-ft-ema

| | AUTOENCODER | RFID↓ | GFID↓ | EQUIV. ERROR | |
|---|---|---|---|---|---|
| | | | | $R(\theta)$↓ | $S(s)$↓ |
| CONT. | SD-VAE | 0.90 | 43.8 | 0.89 | 0.69 |
| | + EQ-VAE (ours) | 0.82 | 34.1 | 0.56 | 0.43 |
| | SDXL-VAE | 0.67 | 46.0 | 1.25 | 0.97 |
| | + EQ-VAE (ours) | 0.65 | 35.9 | 0.65 | 0.35 |
| | SD3-VAE | 0.20 | 58.9 | 0.51 | 0.16 |
| | + EQ-VAE (ours) | 0.19 | 54.0 | 0.37 | 0.11 |
| | SD-VAE-16 | 0.87 | 64.1 | 0.95 | 0.85 |
| | + EQ-VAE (ours) | 0.82 | 49.7 | 0.39 | 0.17 |
| DISC. | VQ-GAN | 7.94 | 6.8 | 1.35 | 1.22 |
| | + EQ-VAE (ours) | 7.54 | 5.9 | 0.64 | 0.55 |

Table 1: **Comparison of Autoencoders with and without EQ-VAE.** We evaluate reconstruction quality, equivariance errors (defined in Appendix C), and generative performance for continuous (SD-VAE, SDXL-VAE, SD3-VAE) and discrete (VQ-GAN) autoencoders, with and without EQ-VAE. Generative FID (GFID) is measured using DiT-B for continuous VAEs and MaskGIT for VQ-GAN. Our approach reduces reconstruction RFID and equivariance errors while enhancing generative performance (GFID). For additional reconstruction metrics see Table 12.

set. To distinguish reconstruction and generation FID, we write GFID and RFID, respectively. To quantify the effectiveness of EQ-VAE we further measure the equivariance error (see Appendix C).

### 4.2. Equivariance-regularized VAEs

We begin our experimental analysis by demonstrating the versatility of EQ-VAE, showing that it seamlessly adapts to both continuous and discrete autoencoders.

**Continuous Autoencoders** We integrate our EQ-VAE regularization into established continuous autoencoders with varying latent dimensions. Namely, SD-VAE, SD-VAE-16, (Rombach et al., 2022), SDXL-VAE (Podell et al., 2024), and SD3-VAE (Esser et al., 2024). To evaluate the effect of the regularization on generative performance we train DiT-B models on the latent codes before and after our regularization. We present our results in Table 1. We observe that our simple objective effectively reduces the equivariance error for all autoencoders. Further, EQ-VAE maintains the original autoencoders' reconstruction fidelity while consistently delivering significant improvements in generative performance. The results hint that there is a correlation between the generative performance (GFID) and the reduction in equivariacne error. Notably, for SD-VAE, SDXL-VAE and SD-VAE-16, our regularization significantly boosts generative performance. For SD3-VAE, although the reduction in equivariance error is relatively modest, it still results in a GFID improvement.

| MODEL | #PARAMS | ITER. | GFID↓ |
|---|---|---|---|
| DiT-B/2 | 130M | 400K | 43.5 |
| w/ EQ-VAE (ours) | 130M | 400K | 34.1 |
| SiT-B/2 | 130M | 400K | 33.0 |
| w/ EQ-VAE (ours) | 130M | 400K | 31.2 |
| DiT-XL/2 | 675M | 400K | 19.5 |
| w/ EQ-VAE (ours) | 675M | 400K | 14.5 |
| SiT-XL/2 | 675M | 400K | 17.2 |
| w/ EQ-VAE (ours) | 675M | 400K | 16.1 |
| DiT-XL/2 | 675M | 7M | 9.6 |
| w/ EQ-VAE (ours) | 675M | 1.5M | 8.8 |
| SiT-XL/2+REPA | 675M | 4M | 5.9 |
| w/ EQ-VAE (ours) | 675M | 1M | 5.9 |

Table 2: **GFID Comparisons.** GFID scores on ImageNet $256 \times 256$ for DiT, SiT, and REPA trained with either SD-VAE-FT-EMA or our EQ-VAE. No classifier-free guidance (CFG) is used. EQ-VAE consistently enhances both generative performance and training efficiency across all generative models.

| MODEL | EPOCH | GFID↓ | IS↑ |
|---|---|---|---|
| MaskGIT | 300 | 6.19 | 182.1 |
| MaskGIT[†] | 300 | 6.80 | 214.0 |
| w/ EQ-VAE (ours) | 130 | 6.80 | 188.1 |
| w/ EQ-VAE (ours) | 300 | 5.91 | 228.8 |

Table 3: **Boosting Masked Generative Modeling.** Comparison of GFID and IS on ImageNet $256 \times 256$ for MaskGIT (Chang et al., 2022) and its open-source PyTorch reproduction[†] (Besnier & Chen, 2023), trained with either VQ-GAN or our EQ-VAE. EQ-VAE accelerates training by more than $\times 2$ (130 vs. 300 epochs), highlighting EQ-VAE can be effectively applied to vector-quantized autoencoders.

**Discrete Autoencoders** To investigate if EQ-VAE can be applied to discrete autoencoders, we experiment on VQ-GAN (Esser et al., 2021) and validate the effectiveness of our regularization on the masked image modeling framework MaskGIT (Chang et al., 2022). In Table 1, we show that EQ-VAE is effective in the discrete case, reducing the equivariance error as well as improving the generative performance from 6.8 to 5.9 in GFID.

### 4.3. Boosting Generative Image Models

By applying EQ-VAE to both continuous and discrete autoencoders, we enhance the performance of state-of-the-art generative models, including DiT a pure transformer diffusion model, SiT that employs continuous flow-based modeling, REPA a recent approach aligning transformer representations with self-supervised features and MaskGIT a well-established masked generative model.

**DiT & SiT** As demonstrated in Table 2, our regularization approach yields significant improvements across

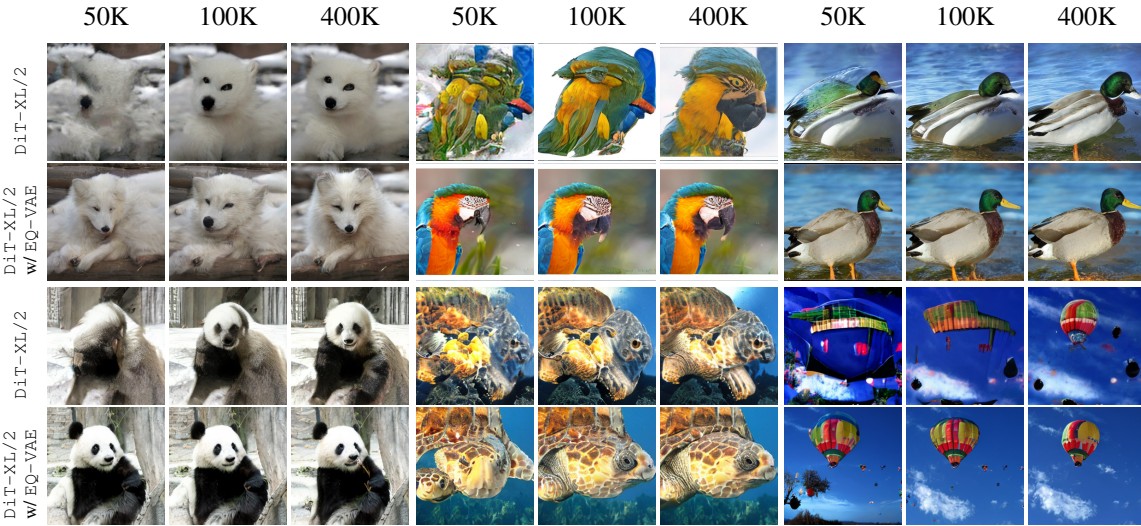

Figure 4: **EQ-VAE accelerates generative modeling.** We compare results from two `DiT-XL/2` models at 50K, 100K, and 400K iterations, one trained with `SD-VAE-FT-EMA` **(top)** and with `EQ-VAE` **(bottom)**. The same noise and number of sampling steps are used for both models, without classifier-free guidance. Our approach delivers faster improvements in image quality, demonstrating accelerated convergence.

both `DiT-B` and `DiT-XL` models. Specifically, training `DiT-XL/2` on the regularized latent distribution achieves GFID 14.5 at 400K iterations, compared to 19.5 without regularization. Notably, by 1.5M iterations, `DiT-XL/2` trained with `EQ-VAE` achieves GFID 8.8, outperforming the `DiT-XL/2` model trained with `SD-VAE-FT-EMA` even at 7M iterations. The speed-up provided by `EQ-VAE` can be qualitatively observed in Figure 4. Moreover, in Table 2, we show that `SiT` models can also benefit from the regularized latent distribution of `EQ-VAE`, improving GFID from 17.2 to 16.1 at 400K steps.

**REPA** We show that our regularization (which is performed in the first stage of latent generative modeling) is complementary to REPA, thus leading to further improvements in convergence and generation performance. Specifically, training REPA (`SiT-XL-2`) with our `EQ-VAE` reaches 5.9 GFID in 1M instead of 4M iterations. Thus, the regularized latent distribution of `EQ-VAE` can make the convergence of REPA ×4 faster (Figure 1). *This is striking because REPA was shown to already significantly speed-up the convergence of diffusion models.*

**MaskGIT** As shown in Table 3, `MaskGIT` trained with our `EQ-VAE` converges twice as fast reaching 6.80 GFID in 130 epochs, instead of 300. Furthermore, by epoch 300 it reaches 5.91 GFID surpassing the performance reported in both (Besnier & Chen, 2023) and (Chang et al., 2022).

**Comparison with state-of-the-art generative models** To further demonstrate how `EQ-VAE` accelerates the learning process, we compare it with recent diffusion methods using classifier-free guidance. Notably, as shown in Table 4,

| MODEL | EPOCHS | GFID↓ | sFID↓ | IS↑ | PRE.↑ | REC.↑ |
|---|---|---|---|---|---|---|
| LDM | 200 | 3.60 | - | 247.7 | 0.87 | 0.48 |
| MaskDiT | 1600 | 2.28 | 5.67 | 276.6 | 0.80 | 0.61 |
| SD-DiT | 480 | 3.23 | - | - | - | - |
| SiT-XL/2 | 1400 | 2.06 | 4.50 | 270.3 | 0.82 | 0.59 |
| DiT-XL/2 | 1400 | 2.27 | 4.60 | 278.2 | 0.83 | 0.57 |
| DiT-XL/2 [†] | 1400 | 2.47 | 5.18 | 276.1 | 0.82 | 0.57 |
| + EQ-VAE (ours) | 300 | 2.37 | 4.78 | 277.3 | 0.82 | 0.57 |
| REPA* | 800 | 1.42 | 4.70 | 305.7 | 0.80 | 0.65 |
| + EQ-VAE * (ours) | 200 | 1.70 | 5.13 | 283.0 | 0.79 | 0.62 |

Table 4: **Comparison on ImageNet 256×256 with CFG.** † indicates that the used autoencoder is the original `SD-VAE` (instead of `SD-VAE-FT-EMA`). REPA uses `SiT-XL/2`. * denotes that guidance interval (Kynkäänniemi et al., 2024) is applied.

`DiT-XL/2` with `EQ-VAE` reaches 2.37 GFID in just 300 epochs, matching the performance of `DiT-XL/2` trained with `SD-VAE` or `SD-VAE-FT-MAE`. Even when combining `EQ-VAE` with the state-of-the-art approach REPA, we are able to achieve comparable results with standard REPA while using ×4 less training compute (200 vs 800 epochs).

### 4.4. Analysis

**Spatial transformations ablation** We begin the analysis of our method by ablating the effect of our equivariance regularization on generative performance with each spatial transformation to understand their respective impact. We consider isotropic $S(s, s)$ or anisotropic $S(s_x, s_y)$ scaling, rotations $R(\theta)$, and combined transformations. We then

| AUTOENCODER | $\tau$ | GFID↓ | RFID↓ | ID |
|---|---|---|---|---|
| SD-VAE | - | 43.5 | 0.90 | 62.2 |
| + EQ-VAE | $R(\theta)$ | 41.2 | 0.73 | 57.9 |
| + EQ-VAE | $S(s, s)$ | 35.8 | 0.78 | 41.0 |
| + EQ-VAE | $R(\theta) \cdot S(s, s)$ | 34.1 | 0.82 | 39.4 |
| + EQ-VAE | $R(\theta) \cdot S(s_x, s_y)$ | 33.2 | 0.92 | 38.9 |

Table 5: **Spatial Transformation Ablation in EQ-VAE.** We measure GFID, RFID, and intrinsic dimension (ID) for latents regularized via rotations, isotropic scaling, anisotropic scaling, and combinations. Combining transformations lowers ID and enhances generative performance, though anisotropic scaling can slightly degrade reconstruction.

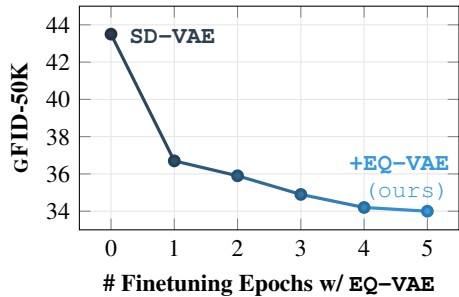

Figure 5: **Rapid Improvement via EQ-VAE Fine-tuning.** Even a single epoch of EQ-VAE fine-tuning significantly improves generative modeling performance, reducing GFID from 43.5 to 36.7. Generative modeling with DiT-B/2.

train a DiT-B/2 on each latent distribution. In Table 5, we observe that encouraging scale equivariance has a significant impact on generative performance. Furthermore, rotation equivariance is also beneficial in generation performance. Combining transformations yields further improvement, demonstrating their complementary effects. While anisotropic scaling yields a better generative performance since the regularization is more aggressive, it negatively impacts reconstruction quality. Thus, our EQ-VAE default setting uses combinations of rotations and isotropic scaling.

**Latent space complexity and generative performance** To better understand the impact of our regularization on the complexity of the latent manifold, we measure its Intrinsic Dimension (ID). The ID represents the minimum number of variables needed to describe a data distribution (Bennett, 1969). Notably, in Table 5, we observe a correlation between the intrinsic dimension of the latent manifold and the resulting generative performance. This suggests that the regularized latent distribution becomes simpler to model, further validating the effectiveness of our approach. This reduction in the complexity of latent representations can also be qualitatively observed in Figure 1 (left). For further details on ID, see Appendix B.

| AUTOENCODER | GFID↓ | RFID↓ |
|---|---|---|
| SD-VAE (Rombach et al., 2022) | 43.8 | 0.90 |
| SD-VAE-FT-EMA (Rombach et al., 2022) | 43.5 | 0.73 |
| SD-VAE† | 43.5 | 0.81 |
| EQ-VAE | 34.1 | 0.82 |

Table 6: **Additional Training vs. Equivariance Regularization.** Comparing various fine-tuning strategies for SD-VAE confirms that EQ-VAE 's improvements stem from equivariance regularization. † Denotes additional training with the standard objective (Eq. (1)) for 5 epochs.

**How many epochs does EQ-VAE need to enhance generation?** To demonstrate how quickly our objective regularizes the latent distribution, we conduct an ablation study by varying the number of fine-tuning epochs. We train a DiT-B/2 model on the resulting latent distribution of each epoch and present the results in Figure 5. Notably, even with a single epoch (10K steps) of fine-tuning, the GFID drops from 43.5 to 36.7, highlighting the rapid refinement our objective achieves. For context, SD-VAE-FT-EMA has been fine-tuned for 300K steps.

**The enhancement in generative performance is not a result of the additional training** To verify that the improvement in generative performance stems from our equivariance regularization (Eq. (5)) rather than additional training, we compare EQ-VAE with SD-VAE† in Table 6. SD-VAE† is obtained by fine-tuning SD-VAE for five extra epochs using only the original objective (Eq. (1)). The results show that this additional training has a negligible effect on generative performance, whereas EQ-VAE leads to a significant improvement. Similarly, SD-VAE-EMA-FT, derived from SD-VAE, has minimal impact on the GFID score, further underscoring the effectiveness of EQ-VAE.

## 5. Conclusion

In this work, we argue that the structure of latent representations produced by the autoencoder is crucial for the convergence speed and performance of latent generative models. We observed that latent representations of established autoencoders are not equivariant under simple spatial transformations. To address this, we introduce EQ-VAE, a simple modification to the autoencoder's training objective. We empirically demonstrated that fine-tuning pre-trained autoencoders with EQ-VAE for just a few epochs, is enough to reduce the equivariance error and significantly boost the performance of latent generative models while maintaining their reconstruction capability. We believe that our work introduces several promising future directions, particularly in exploring the theoretical and empirical relationship between the geometry of the latent distribution and the performance of latent generative models.

## Impact Statement

This paper presents work whose goal is to advance the field of machine learning in general and image synthesis in particular. There are many potential societal consequences of our work, none of which we feel must be specifically highlighted here.

**Acknowledgements** This work has been partially supported by project MIS 5154714 of the National Recovery and Resilience Plan Greece 2.0 funded by the European Union under the NextGenerationEU Program.

Hardware resources were granted with the support of GR-NET. Also, this work was performed using HPC resources from GENCI-IDRIS (Grants 2024-AD011012884R3).

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

# Contents

# A. Additional Ablations

## A.1. Implicit vs Explicit Equivariance Regularization

Here, we provide an analysis of the design choice of our objective. We aim to design an objective that reduces the equivariance error of the encoder while avoiding mode collapse and preserving reconstruction performance. For each objective investigated, we finetune `SD-VAE` and evaluate the effect on generative performance by training a `DiT-B/2` on the resulting latent distribution. Initially, we perform fine-tuning with the standard objective along with the explicit loss in (Eq. (4)): $\mathcal{L}_{VAE} + \lambda\mathcal{L}_{\text{explicit}}$ and set $\lambda = 0.1$. We further experiment with adding a stop-gradient (sg) in the $\mathcal{E}(\tau \circ \mathbf{x})$ term in $\mathcal{L}_{\text{explicit}}$. In Table 7, we observe that using $\mathcal{L}_{\text{explicit}}$ successfully reduces the equivariance error for both rotation and scaling transformations. However, both reconstruction and generative performance degrade severely, indicating a mode collapse in the latent space.

| Loss | GFID↓ | RFID↓ | Equivariance Error | |
| --- | --- | --- | --- | --- |
| | | | $R(\theta)\downarrow$ | $S(s)\downarrow$ |
| SD-VAE | 43.5 | 0.90 | 0.93 | 0.80 |
| w/ explicit | 141.3 | 117.93 | 0.32 | 0.11 |
| w/ explicit + sg | 134.7 | 109.25 | 0.35 | 0.13 |
| w/ implicit (ours) | 34.1 | 0.82 | 0.49 | 0.15 |

Table 7: **Implicit vs. Explicit Equivariance Regularization.** Comparing `SD-VAE` along with explicit vs implicit regularization objectives shows that explicit regularization drastically lowers equivariance errors but triggers mode collapse, while implicit regularization enhances significantly the generative performance.

## A.2. Regularization Strength

We evaluate the impact of hyperparameter $p_\alpha$ which controls the strength of our regularization in Table 8.

| Autoencoder | $p_\alpha$ | GFID ↓ | RFID ↓ |
| --- | --- | --- | --- |
| EQ-VAE | 0.3 | 35.4 | 0.78 |
| EQ-VAE | 0.7 | 34.4 | 0.88 |
| EQ-VAE | 0.5 | 34.1 | 0.82 |

Table 8: **Ablation on regularization strength.** We perform two experiments, with lower ($p_\alpha = 0.7$) and higher ($p_\alpha = 0.3$) regularization strength. We observe that our method is relatively robust to choices of $p_\alpha$. We highlight the setting used throughout all our experiments.

# B. Details on the Intrinsic Dimension Estimation.

Several recent works (Valeriani et al., 2023; Kvinge et al., 2023; Cheng et al., 2023) have utilized ID to measure the complexity of latent representations in deep learning modeling. Further Pope et al. (2021) has demonstrated a strong correlation between a dataset's relative difficulty and its ID. We compute the ID of the latent representations using the TwoNN estimator (Facco et al., 2017), which relies solely on the distances between each point and its two nearest neighbors. In practice, the TwoNN estimator can be affected by noise, which typically leads to an overestimation of the ID. Nevertheless, it is a robust tool to evaluate *relative* complexity and has been used effectively to analyze representations in deep neural networks (Valeriani et al., 2023). We adopt the TwoNN implementation of DADAPy (Glielmo et al., 2022).

# C. Details on Evaluation Metrics

## C.1. Generation Metrics

We follow the setup and use the same reference batches of ADM (Nichol & Dhariwal, 2021) for evaluation, utilizing their official implementation[2]. We use NVIDIA A100 GPUs for our evaluation. We briefly explain each metric used for the evaluation.

- **FID** (Heusel et al., 2017) quantifies the feature distance between the distributions of two image datasets by leveraging the Inception-v3 network (Szegedy et al., 2016). The distance is calculated based on the assumption that both feature distributions follow multivariate Gaussian distributions.

- **sFID** (Nash et al., 2021) computes FID with intermediate spatial features of the Inception-v3 network, to capture spatial distribution of the generated images

- **IS** (Salimans et al., 2016) measures a KL-divergence between the original label distribution and the distribution of Inception-v3 network's logits after the softmax normalization.

- **Precision and Recall** (Kynkäänniemi et al., 2019) are the fraction of realistic images and the fraction of training data covered by generated data respectivly.

---

[2]https://github.com/openai/guided-diffusion/tree/main/evaluations

## C.2. Reconstruction Metrics

We evaluate reconstruction on the validation set of Imagenet which contains 50K images. We provide a description of each metric used for the reconstruction evaluation.

- **PSNR** measures the quality of reconstructed images by comparing the maximum possible signal power to the level of noise introduced during reconstruction. Expressed in decibels (dB).

- **SSIM** (Wang et al., 2004) assesses the similarity between two images by evaluating their structural information, luminance, and contrast.

- **LPIPS** (Zhang et al., 2018) evaluates the perceptual similarity between two images by comparing their deep feature representations using VGG (Simonyan, 2014)

## C.3. Equivariance Error

To quantify the effectiveness of $\texttt{EQ-VAE}$ at constraining the latent representations of the autoencoders to equivary under scale and rotation transformation we measure the equivariance error. Similar to (Sosnovik et al., 2020) we define the equivariance error as follows: $\Delta_{eq}^{\mathcal{T}} = \frac{1}{|\mathcal{T}| \cdot N} \sum_{|\mathcal{T}|} \sum_{N} \|\tau \circ \mathcal{E}(\mathbf{x}) - \mathcal{E}(\tau \circ \mathbf{x})\|_2^2 \,/\, \|\mathcal{E}(\tau \circ \mathbf{x})\|_2^2$ where N = 50K in the number of samples in ImageNet validation and $\mathcal{T}$ is the set of transformations considered. We conduct our evaluation with $\mathcal{T}_r = \{\frac{\pi}{2}, \pi, \frac{3\pi}{2}\}$ for rotations and $\mathcal{T}_s = \{0.25, 0.50, 0.75\}$ for scale.

# D. Detailed Benchmarks

## D.1. Detailed generative performance

We provide a detailed evaluation of all the generative models presented in the main paper, including additional metrics and training iterations. Specifically, Table 9 details the performance of the $\texttt{DiT-XL/2}$ and $\texttt{SiT-XL/2}$ models, while Table 10 presents results for the REPA ($\texttt{SiT-XL/2}$) models trained with both $\texttt{SD-VAE-FT-EMA}$ (as reported in the respective papers) and $\texttt{EQ-VAE}$. Additionally, Table 11 provides results for $\texttt{MaskGIT}$ models trained using $\texttt{VQ-GAN}$ and $\texttt{EQ-VAE}$. For all models, we use the evaluation metrics originally reported in the original publications.

| MODEL | #ITERS. | FID↓ | sFID↓ | IS↑ | PREC.↑ | REC.↑ |
|---|---|---|---|---|---|---|
| DiT-XL/2 (Peebles & Xie, 2023) | 400K | 19.5 | 6.5 | 77.5 | 0.60 | 0.60 |
| w/ EQ-VAE | 50K | 73.6 | 13.1 | 34.5 | 0.50 | 0.37 |
| w/ EQ-VAE | 100K | 39.9 | 6.8 | 62.2 | 0.60 | 0.53 |
| w/ EQ-VAE | 200K | 22.8 | 5.9 | 73.6 | 0.61 | 0.62 |
| w/ EQ-VAE | 400K | 14.5 | 5.6 | 81.5 | 0.63 | 0.66 |
| SiT-XL/2 (Ma et al., 2024) | 400K | 17.2 | 5.1 | 76.5 | 0.64 | 0.63 |
| w/ EQ-VAE | 50K | 76.1 | 38.4 | 15.2 | 0.50 | 0.37 |
| w/ EQ-VAE | 100K | 41.3 | 10.9 | 30.9 | 0.60 | 0.53 |
| w/ EQ-VAE | 200K | 24.9 | 6.4 | 54.6 | 0.61 | 0.62 |
| w/ EQ-VAE | 400K | 16.1 | 4.2 | 79.7 | 0.64 | 0.66 |

Table 9: **Detailed evaluation** for $\texttt{DiT-XL/2}$ and $\texttt{SiT-XL/2}$ models. All results are reported without classifier-free guidance (CFG = 1.0).

| MODEL | #ITERS. | FID↓ | sFID↓ | IS↑ | PREC.↑ | REC.↑ |
|---|---|---|---|---|---|---|
| REPA (Yu et al., 2025) | 50K | 52.3 | 31.2 | 24.3 | 0.45 | 0.53 |
| w/ EQ-VAE | 50K | 48.7 | 26.3 | 27.6 | 0.44 | 0.53 |
| REPA (Yu et al., 2025) | 100K | 19.4 | 6.1 | 67.4 | 0.64 | 0.610 |
| w/ EQ-VAE | 100K | 18.7 | 5.4 | 67.8 | 0.65 | 0.59 |
| REPA (Yu et al., 2025) | 200K | 11.1 | 5.1 | 100.4 | 0.69 | 0.64 |
| w/ EQ-VAE | 200K | 10.7 | 5.1 | 103.5 | 0.70 | 0.62 |
| REPA (Yu et al., 2025) | 400K | 7.9 | 5.1 | 122.6 | 0.70 | 0.65 |
| w/ EQ-VAE | 400K | 7.5 | 5.0 | 128.8 | 0.71 | 0.63 |

Table 10: **Detailed evaluation** on REPA (SiT-XL/2) models. All results are reported without classifier-free guidance (CFG = 1.0)

| MODEL | EPOCHS | FID↓ | IS↑ | PREC.↑ | REC.↑ |
|---|---|---|---|---|---|
| MaskGIT (Chang et al., 2022) | 300 | 6.2 | 182.1 | 0.80 | 0.51 |
| MaskGIT* (Besnier & Chen, 2023) | 300 | 6.8 | 214.0 | 0.82 | 0.51 |
| w/ EQ-VAE | 50 | 11.1 | 116.1 | 0.73 | 0.52 |
| w/ EQ-VAE | 100 | 7.6 | 167.5 | 0.78 | 0.53 |
| w/ EQ-VAE | 200 | 6.0 | 211.8 | 0.79 | 0.55 |
| w/ EQ-VAE | 300 | 5.9 | 228.7 | 0.80 | 0.55 |

Table 11: **Detailed evaluation** on MaskGIT models. We use the open-source PYTORCH reproduction for our experiments. All results are reported without classifier-free guidance (CFG = 3.0)

## D.2. Detailed reconstruction performance

We detail in Table 12 the reconstruction evaluation metrics of each autoencoder with and without EQ-VAE regularization.

| | AUTOENCODER | rFID↓ | PSNR↑ | LPIPS↓ | SSIM↑ |
|---|---|---|---|---|---|
| | SD-VAE | 0.90 | 25.82 | 0.146 | 0.71 |
| | w/ EQ-VAE (ours) | 0.82 | 25.95 | 0.141 | 0.72 |
| | SDXL-VAE | 0.67 | 27.36 | 0.121 | 0.76 |
| | w/ EQ-VAE (ours) | 0.65 | 27.48 | 0.118 | 0.76 |
| CONT. | SD3-VAE | 0.20 | 31.27 | 0.060 | 0.87 |
| | w/ EQ-VAE (ours) | 0.19 | 31.06 | 0.061 | 0.87 |
| | SD-VAE-16 | 0.87 | 24.67 | 0.161 | 0.61 |
| | w/ EQ-VAE (ours) | 0.82 | 25.21 | 0.152 | 0.69 |
| DISC. | VQ-GAN | 7.94 | 19.41 | 0.540 | 0.54 |
| | w/ EQ-VAE (ours) | 7.54 | 19.61 | 0.510 | 0.56 |

Table 12: **Comparison of Autoencoders with and without EQ-VAE**. Additional reconstruction evaluation metrics.

# E. Specifications of Autoencoder Models

| AUTOENCODER | $\mathcal{L}_{reg}$ | DATASET | $c$ | $f$ |
|---|---|---|---|---|
| SD-VAE (Rombach et al., 2022) | KL | OpenImages | 4 | 8 |
| SD-VAE-FT-EMA (Rombach et al., 2022) | KL | OpenImages + Laion Aesthetics | 4 | 8 |
| SD-VAE-16 (Rombach et al., 2022) | KL | OpenImages | 16 | 16 |
| SDXL-VAE (Podell et al., 2024) | KL | - | 4 | 8 |
| VQ-GAN (Esser et al., 2021) | VQ | ImageNet | 256 | 16 |

Table 13: **Specifications of Autoencoders**. We provide additional information for the autoencoders used in our experiments regarding their original training dataset, latent channels $c$, and compression rate $f$.

# F. Latent Generative Models

Here we provide a brief description of the latent generative models, mentioned in the main paper:

- MaskGIT (Chang et al., 2022) utilizes a bidirectional transformer decoder to synthesize images by iteratively predicting masked visual tokens produced by a VQ-GAN (Esser et al., 2021).
- LDM (Rombach et al., 2022) proposes latent diffusion models, modeling the image distribution in a compressed latent space produced by a KL- or VQ-regularized autoencoder.
- DiT (Yao et al., 2024) proposes a pure transformer backbone for training diffusion models and incorporates AdaIN-zero modules.
- MaskDiT (Zheng et al., 2023) trains diffusion transformers with an auxiliary mask reconstruction task.
- SD-DiT (Zhu et al., 2024b) extends the MaskDiT architecture by incorporating a discrimination objective using a momentum encoder.
- SiT (Ma et al., 2024) improves diffusion transformer training by moving from discrete diffusion to continuous flow-based modeling.
- REPA (Yu et al., 2025) aligns the representations of diffusion transformer models to the representations of self-supervised models.

## G. Additional Qualitative Results

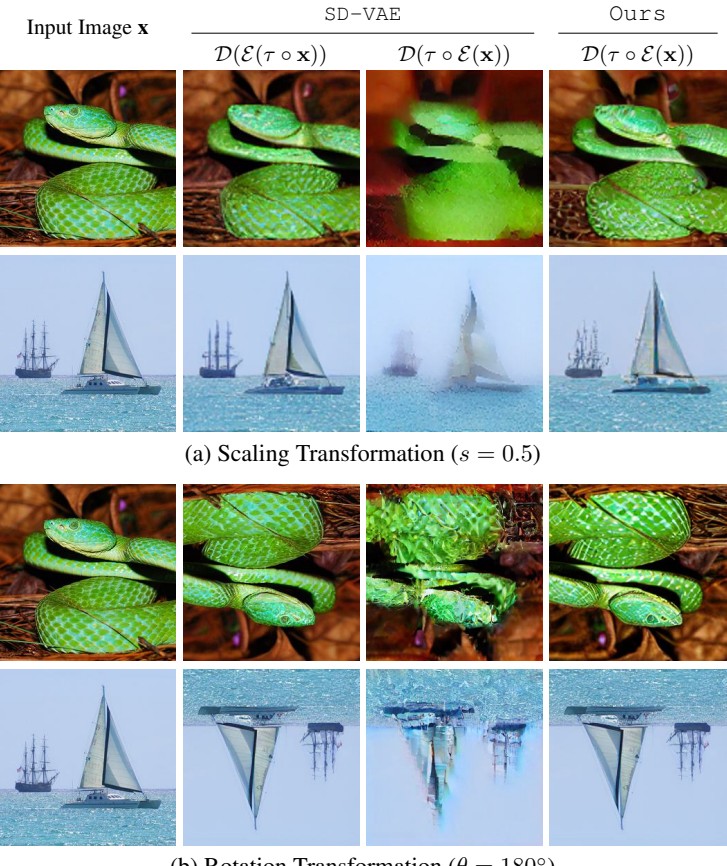

(a) Scaling Transformation ($s = 0.5$)

(b) Rotation Transformation ($\theta = 180°$)

Figure 6: **Latent Space Equivariance.** Reconstructed images using SD-VAE (Rombach et al., 2022) and EQ-VAE when applying transformations $\tau$ to the input images ($\mathcal{D}(\mathcal{E}(\tau \circ \mathbf{x}))$) versus directly to the latent representations ($\mathcal{D}(\tau \circ \mathcal{E}(\mathbf{x}))$). We present results for scaling and rotation transformations $\tau$. Our approach preserves reconstruction quality under latent transformations, whereas SD-VAE exhibits significant degradation.

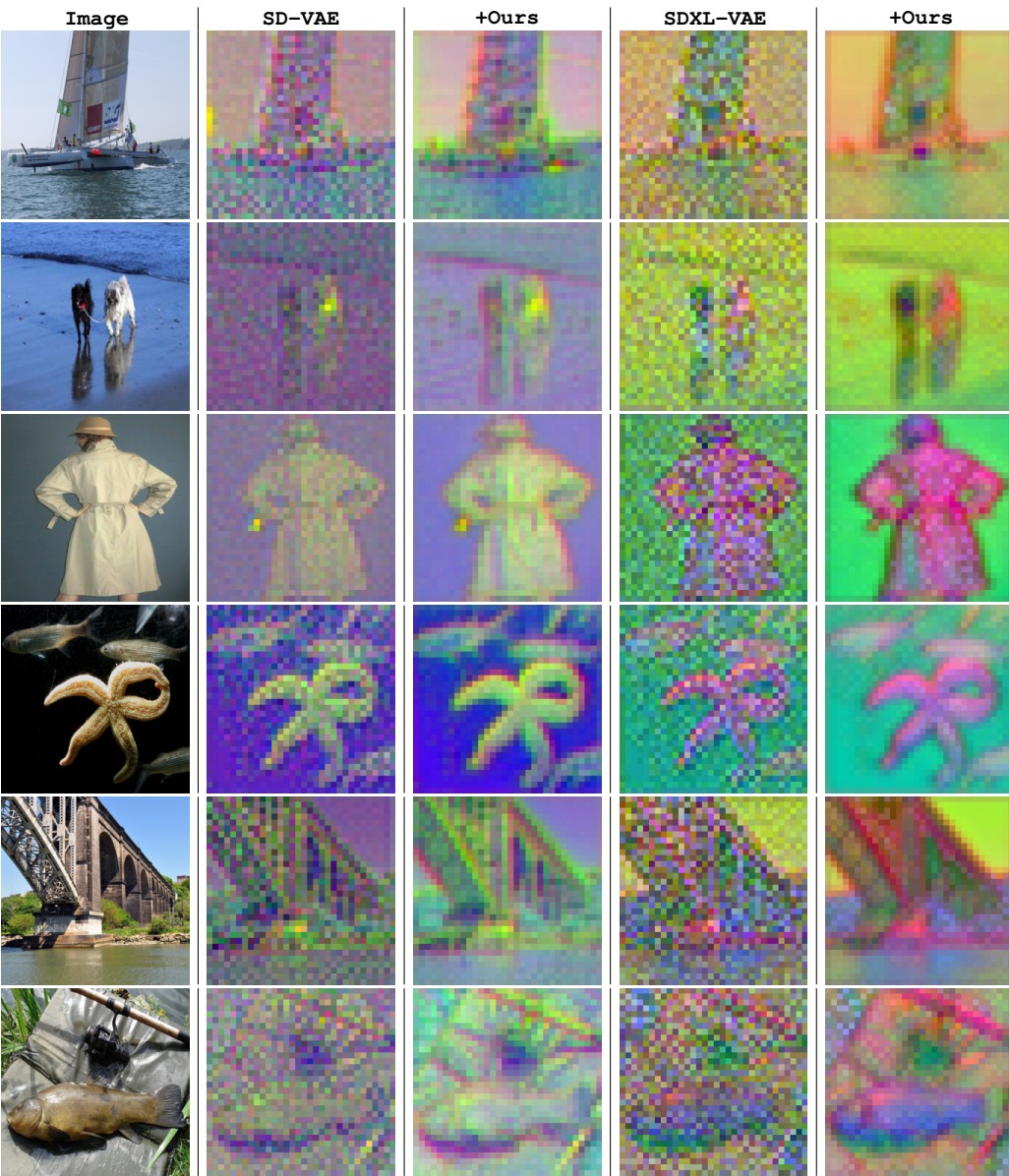

Figure 7: **Additional comparisons of latent representations** across different VAE models. `EQ-VAE` (+Ours) produces smoother latent representations for both `SD-VAE` and `SDXL-VAE`.

Class label = "panda" (388)

Class label = "golden retriever" (207)

Class label = "macaw" (88)

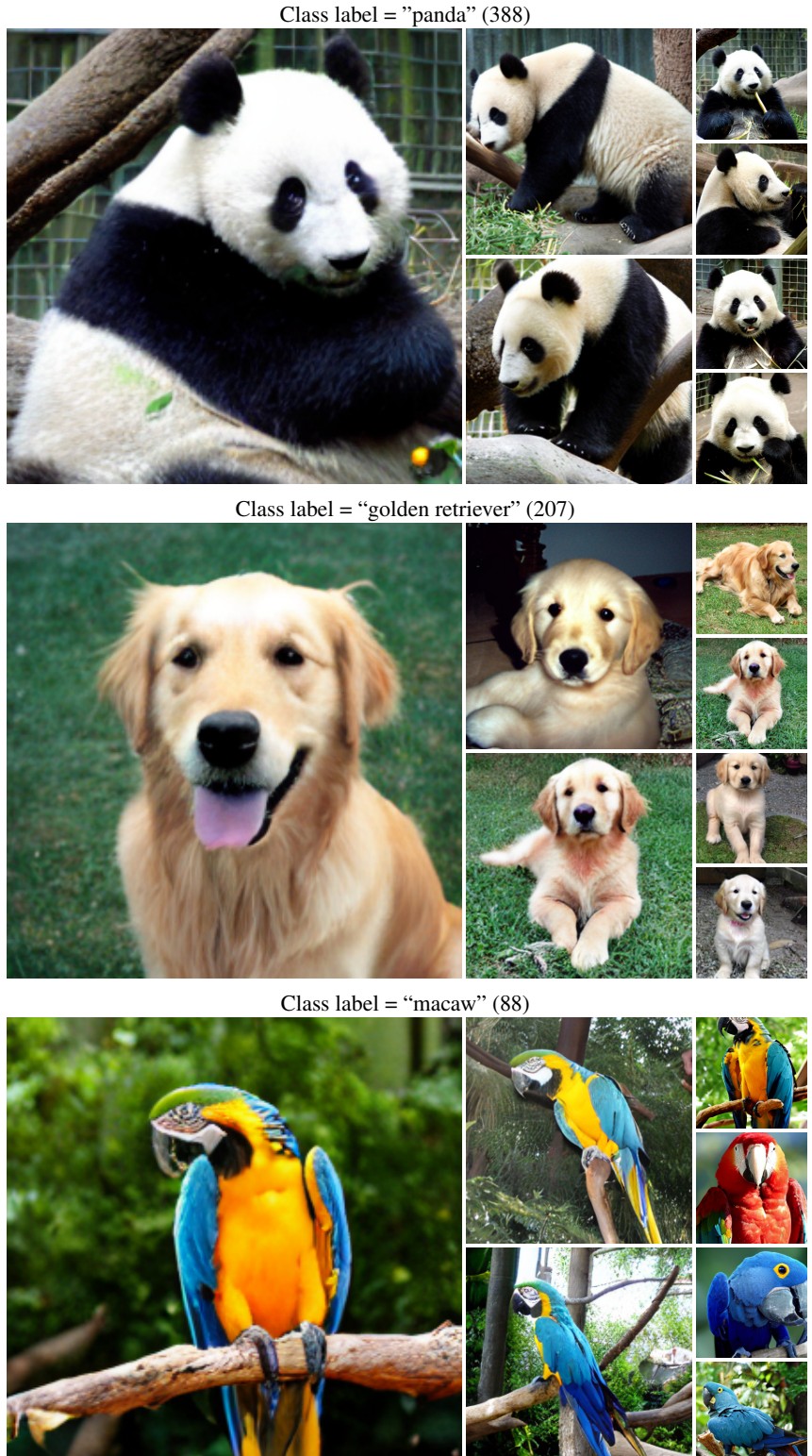

Figure 8: **Uncurated samples $256 \times 256$ `DiT-XL/2 /w EQ-VAE`.** Classifier-free guidance scale = 4.0.

