# OpenReview forum: "EQ-VAE: Equivariance Regularized Latent Space for Improved Generative Image Modeling"
_ICML.cc/2025/Conference — ICML 2025 poster_

### Official Review · Reviewer_k5zo · 2025-03-10

**Overall Recommendation:** 3

**Summary:**

This paper addresses the limitations of existing latent generative models, which often lack equivariance to semantic-preserving transformations like scaling and rotation. To overcome this challenge, the authors propose EQ-VAE, a regularization technique that enforces equivariance in the latent space, simplifying it while preserving reconstruction quality. EQ-VAE enhances the performance of various state-of-the-art generative models and is compatible with both continuous and discrete autoencoders, providing a versatile improvement for a range of latent generative frameworks.

**Claims And Evidence:**

The authors provide empirical results demonstrating the effectiveness of their method, EQ-VAE, in improving the performance of both continuous and discrete autoencoders. Specifically, they mention significant speedups in training times and improvements in downstream model performance, as measured by FID scores.

**Essential References Not Discussed:**

There is no significant related work that is not discussed.

**Experimental Designs Or Analyses:**

By comparing EQ-VAE against baseline models like SD-VAE and SD-VAE-EMA-FT, the authors convincingly demonstrate that improvements in generative performance are attributed to their proposed method rather than just additional training. No significant issues were noted in the designs or analyses.

**Methods And Evaluation Criteria:**

The use of benchmark datasets and FID scores as evaluation criteria is appropriate for assessing the performance of image synthesis methods. These metrics provide a clear way to quantify improvements in image quality and training efficiency, making the evaluation relevant and meaningful for the context of latent generative modeling.

**Other Comments Or Suggestions:**

No other suggestions.

**Other Strengths And Weaknesses:**

Strengths

The paper addresses critical challenges in latent generative modeling, specifically the trade-offs between reconstruction quality and generative performance. By proposing a solution that enhances both aspects, the work has significant implications for advancing state-of-the-art generative models.

Weaknesses

The experimental results in Table 2 show that the improvement of the baseline by using EQ-VAE is relatively small. For example, SiT-B/2, 400K. The experimental result of REPA under the same configuration is about 25. Does this mean that the improvement of the generated results by this method is relatively small?

**Questions For Authors:**

No other questions.

**Relation To Broader Scientific Literature:**

The paper builds upon the foundational work of latent variable models, particularly in the context of variational autoencoders (VAEs). By applying their method to enhance models like DiT and SiT, the authors demonstrate a significant improvement in generative performance.

**Theoretical Claims:**

This paper does not involve theoretical claims.

---

> ### Author Rebuttal · Authors · 2025-03-31
>
> We appreciate your insightful comments and efforts in reviewing our manuscript. Below, we provide our responses to each of your comments:
>
> ---
>
> **W1. Minor improvements in Table 2 compared to REPA.**
>
> We respectfully emphasize that in Table 2 EQ-VAE demonstrates substantial improvements across all models (DiT, SiT, and REPA) in both B and XL configurations. While the improvement of EQ-VAE for SiT-B/2 may appear relatively small compared to REPA, this comparison requires important contextual considerations:
>
> - REPA is a distillation strategy applied directly in the generative diffusion stage, which inherently leads to improved performance, particularly with the powerful pre-trained visual encoder (DINOv2r) it leverages.
> - In contrast, our method, EQ-VAE, is applied in the autoencoding stage and does not rely on any external model. Instead, it regularizes the latent space, resulting in enhanced performance for generative modeling.
>
> Moreover, as shown in Table 2 and Figure 1 (right), *EQ-VAE is orthogonal to REPA*, accelerating its convergence by a factor of four. This highlights EQ-VAE’s ability to improve efficiency without sacrificing performance.
>
> Lastly, we note that in our reproduction environment, SiT-B/2 achieved a higher 34.7 FID compared to the 33.0 FID reported in the paper, making the improvement attributed to EQ-VAE slightly larger (34.7 → 31.2 FID).

---

### Official Review · Reviewer_28Pq · 2025-03-13

**Overall Recommendation:** 4

**Summary:**

The paper introduces EQ-VAE, a novel variant of autoencoders designed to enhance the performance of latent generative models. The authors first identify that commonly used autoencoders in modern generative models are not equivariant to spatial transformations of the input, such as scaling and rotation. They argue that enforcing this property can lead to improved generation quality. To address this, they propose a new implicit regularization loss that encourages the encoder network to be equivariant with respect to scaling and rotation. Experimental results demonstrate that incorporating this regularization not only accelerates the training of generative models but also improves their performance in certain cases.

**Claims And Evidence:**

The claims are sufficiently supported with various experiments.

**Essential References Not Discussed:**

The paper includes all essential references.

**Experimental Designs Or Analyses:**

The experiments are well-designed, and the evaluation setup is fair across various models and architectures.

**Methods And Evaluation Criteria:**

The method is evaluated using common metrics such as recnostruction FID, generation FID, Inception Score, and LPIPS.

**Other Comments Or Suggestions:**

It would be interesting to explore how EQ-VAE impacts other autoencoders designed to improve the efficiency of SD-VAE, such as Cosmos-Tokenizer [1] and LiteVAE [2].

[1] Agarwal N, Ali A, Bala M, Balaji Y, Barker E, Cai T, Chattopadhyay P, Chen Y, Cui Y, Ding Y, Dworakowski D. Cosmos world foundation model platform for physical ai. arXiv preprint arXiv:2501.03575. 2025 Jan 7.

[2] Sadat S, Buhmann J, Bradley D, Hilliges O, Weber RM. Litevae: Lightweight and efficient variational autoencoders for latent diffusion models. arXiv preprint arXiv:2405.14477. 2024 May 23.

**Other Strengths And Weaknesses:**

### **Strengths**
- The paper is well-presented, making it clear and enjoyable to read.
- Given that latent generative models are the predominant approach for high-resolution image generation, the contributions of this work are likely to have a significant impact on the field.
- While most recent studies focus primarily on the diffusion or generative components of such systems, this paper explores a relatively underexplored area by improving the latent space of the autoencoder.

### **Weaknesses**
- The main weakness of the work is that integrating EQ-VAE into generative models can lead to a performance drop for some models when using classifier-free guidance (e.g., REPA in Table 4).  It would be valuable to assess how much additional training is required to recover the original model's performance. This provides a clearer evaluation of the convergence speed, especially since such models are rarely used without classifier-free guidance.

**Questions For Authors:**

1) What is the reasoning behind selecting only scaling and rotation as the equivariant operations?

2) Does adding the EQ-VAE loss in the context of VQ-VAE introduce any instability or impact due to the additional quantization step?

3) Do you have any intuition about why the performance gap widens after 400k iterations? Specifically, why do different VAEs perform similarly in the early training phase, but diverge later?

**Relation To Broader Scientific Literature:**

While regularization techniques for enforcing equivariance in deep learning models have been explored in recent work, none have directly addressed this issue in the context of generative models. Moreover, no prior study has examined the impact of an equivariant latent space on generation quality. Accordingly, the paper's contributions are well-positioned within the broader literature.

**Theoretical Claims:**

There is no theoretical claim in the paper.

---

> ### Author Rebuttal · Authors · 2025-03-31
>
> We appreciate your insightful comments and efforts in reviewing our manuscript. Below, we provide our responses to each of your comments:
>
> ---
>
> **W1. Results with CFG for converged models.**
>
> We appreciate the reviewer’s concern regarding potential performance degradation when integrating EQ-VAE into generative models with classifier-free guidance (CFG). However, we respectfully clarify that EQ-VAE does not inherently degrade performance.
>
> As shown in Table 4, REPA (trained for 800 epochs) and REPA with EQ-VAE (trained for only 200 epochs) cannot yet be directly compared due to the significant difference in training duration. Notably, DiT-XL/2 with EQ-VAE—trained for just 300 epochs—already outperforms DiT-XL/2† (which uses SD-VAE before EQ-VAE fine-tuning), even though the latter was trained to full convergence (1400 epochs). This suggests that EQ-VAE not only accelerates convergence but may also enhance performance under CFG.
>
> Due to our limited computational resources and short rebuttal timeline, we were unable to train XL models to full convergence. However, we understand the importance of assessing convergence speed in the CFG setting and will include fully converged experiments in the final version of our paper to provide a clearer evaluation.
>
> ---
>
> **S1. EQ-VAE with efficient autoencoders.**
>
> We appreciate the reviewer’s suggestion. Investigating whether our equivariance regularization can be applied to efficient autoencoders such as [1] and [2] is an interesting topic for further exploration. We will include this direction in the Future Work section of our paper.
>
> **Q1. Reasoning behind selecting scaling and rotation.**
>
> The development of scale and rotation equivariant networks [3],[4] has been extensively studied for various image understanding problems.
> This motivated us to explore whether well-established autoencoders utilized in latent generative modeling are equivariant under those basic semantic preserving transformations.
> Our findings, showcased in Figures 2 and 6, revealed that existing architectures lack this property, directly motivating our equivariance regularization approach.
>
> ---
>
> **Q2. VQ-VAE training instability.**
>
> We did not encounter any instability during VQ-VAE fine-tuning.
>
> ---
>
>  **Q3. Performance gap widens after 400k iterations?**
>
> We observe that the performance gap remains consistent across all training iterations. As an example, we present a gFID comparison of REPA without and with EQ-VAE at different training iterations.
>
> | **Iter.** | **REPA (gFID)** | **REPA w/ EQ-VAE (gFID)** |
> |:---------------:|:----------------:|:--------------------------:|
> |          50K   |            52.3 |                     48.7 |
> |         100K   |            19.4 |                     18.7 |
> |         200K   |            11.1 |                     10.7 |
> |         400K   |             7.9 |                      7.5 |
> |           1M   |             6.4 |                      5.9 |
>
> ---
>
> [3] Group equivariant convolutional networks. In ICLR, 2016.
>
> [4] Scaleequivariant steerable networks. In ICLR, 2020.

---

> > ### Comment · Reviewer_28Pq · 2025-04-07
> >
> > I’d like to thank the authors for addressing my questions in the rebuttal. I believe the paper presents valid contributions, and I would like to maintain my score as Accept.

---

### Official Review · Reviewer_JW8k · 2025-03-13

**Overall Recommendation:** 4

**Summary:**

The paper proposes EQ-VAE, a framework that introduces equivariance regularization into the training of autoencoders. By incorporating 2D transformations such as rotation and scaling, the method improves the structure and representation ability of the latent space. As a result, it accelerates the training of generative models and enhances generation quality. The authors demonstrate the effectiveness of EQ-VAE through extensive experiments on both discrete and continuous autoencoders, showing consistent improvements in performance across various generative modeling tasks.

**Claims And Evidence:**

Yes, the claims made in the submission are supported by clear and convincing evidence. The authors demonstrate that their proposed training strategy effectively improves the reconstruction capability of the VAE itself. Furthermore, they provide experimental results showing that, by offering a more expressive latent space through equivariance regularization, the subsequent training of generative models is also improved. This leads to better generation quality and faster convergence. The experiments cover both discrete and continuous autoencoders, and the evaluations are consistent with the claims presented in the paper.

**Essential References Not Discussed:**

No, the paper covers the most relevant prior work related to equivariant representation learning and variational autoencoders. The cited literature provides sufficient context for understanding the key contributions of the paper. I did not identify any essential references that are missing or overlooked.

**Experimental Designs Or Analyses:**

Yes, I checked the soundness and validity of the experimental designs and analyses. The authors conduct thorough comparative experiments on various VAE models, as well as a series of generative models.

**Methods And Evaluation Criteria:**

Yes, the proposed methods and evaluation criteria make sense for the problem at hand. The authors adopt standard datasets commonly used in image generation tasks, such as OpenImages and ImageNet. They also evaluate their method using widely accepted metrics, including FID and sFID, which are appropriate for assessing the quality and diversity of generated images. The experimental setup aligns well with the goals of the paper and provides a fair basis for comparison.

**Other Comments Or Suggestions:**

No

**Other Strengths And Weaknesses:**

Please refer to other sections.

**Questions For Authors:**

No.

**Relation To Broader Scientific Literature:**

The key contributions of the paper are closely related to the broader literature on improving latent space representations in generative models, particularly Variational Autoencoders (VAEs). Prior works have demonstrated that the structure and consistency of the latent space play a critical role in the performance of generative models, both in terms of reconstruction quality and sample generation. This paper builds on these findings by introducing equivariance regularization, ensuring that the latent space behaves consistently under geometric transformations such as rotation and scaling.

Unlike previous approaches that focus on improving VAE expressiveness through architecture changes or better priors, EQ-VAE emphasizes geometric consistency, which has been underexplored in this context. The work is also related to recent efforts in incorporating equivariance into deep learning models, but it uniquely applies this concept to enhance the quality and robustness of latent representations in VAEs, which in turn benefits downstream generative tasks.

**Theoretical Claims:**

Why does the proposed fine-tuning strategy effectively enhance the representational capacity of VAEs and improve the convergence behavior of downstream generative models? While the paper provides intuitive explanations and empirical results to justify these benefits, a more rigorous theoretical analysis or mathematical justification would significantly strengthen the work. For instance, formalizing how equivariant regularization influences the geometry of the latent space, or how it impacts the optimization landscape of generative models, would offer deeper insights beyond empirical validation.

---

> ### Author Rebuttal · Authors · 2025-03-31
>
> We appreciate your insightful comments and efforts in reviewing our manuscript. Below, we provide our responses to each of your comments:
>
> ---
>
> **Theoretical validation**
>
> While we focus on empirical evidence in this work,  we believe that mathematically formalizing the underlying mechanisms behind the success of equivariance regularization in generative models will be an interesting future direction to explore. Our empirical observation that equivariance regularization reduces the intrinsic dimension of the latent manifold—correlating with improvements in generative performance (as shown in Table 5) is an interesting starting point for future theoretical research.
>
> The reviewer’s suggested directions, such as investigating how equivariance regularization influences the geometry of the latent space and its impact on the optimization landscape of generative models, are particularly compelling. We will incorporate these insights into the Future Work section of our paper.

---

### Official Review · Reviewer_bi5M · 2025-03-16

**Overall Recommendation:** 3

**Summary:**

This paper observes that existing autoencoders lack equivariance to semantic-preserving transformations like scaling and rotation, resulting in complex latent spaces that hinder generative performance. Based on this observation, the authors propose to regularize the latent by enforcing equivariance in the latent space, reducing its complexity without degrading reconstruction quality. Experiments on different generative models demonstrate the effectiveness of the proposed method.

## update after rebuttal
Most of my concerns have been addressed and I lean to keep the positive rating.

**Claims And Evidence:**

Yes, the claims made in the submission are supported by clear and convincing evidence.

**Essential References Not Discussed:**

N/A

**Experimental Designs Or Analyses:**

Yes, I have checked the soundness of the experimental designs and analyses. Seems fine to me.

**Methods And Evaluation Criteria:**

Yes, the proposed methods and evaluation criteria make sense for the problem or application at hand.

**Other Comments Or Suggestions:**

N/A

**Other Strengths And Weaknesses:**

Strengths:
1. The idea of exploring the property of latent space is applaudable, which not only brings practical speedup of latent generative models, but provide insights for the design of VAEs.
2. The proposed EQ-VAE can generalize to both continuous and discrete autoencoders with only a few epochs finetuning, and experiments on multiple generative models demonstrate the effectiveness of the proposed method.
3. The paper is well-organized and the writing is clear.

Weaknesses:
1. While the effectiveness of the proposed method has been validated with image generation on ImageNet, its efficacy on text-to-image, which requires larger training sets, remains unverified. The reviewer understands that it would take much larger training resources for T2I experiments, but the translation of the effectiveness to T2I cannot be guaranteed without empirical validation.
2. Although EQ-VAE has shown advantage in reducing the training costs compared to the baseline methods, whether further training could bring additional performance improvement is unclear, that is, whether EQ-VAE just makes the latent generative models converge faster or it could result in better performance of latent generative models with the same training resources as the baselines.
3. While EQ-VAE has compared with baseline VAE in rFID, this metric may be aligned with the training iterations and it is suggested to provide comprehensive comparisons with baseline VAE, finetuned VAE with old objectives and the proposed method with more metrics like PSNR, SSIM and LPIPS.

**Questions For Authors:**

Although the authors mentioned that they only finetune the VAE for 5 epochs on OpenImages, it would be better to have a sense of comparisons between the finetuning costs and the original pre-training costs if applicable. The reason is that different VAEs may train their models with different iterations and batch sizes.

**Relation To Broader Scientific Literature:**

This paper shares similar motivation with prior work like REPA which aims to enhance the training efficiency of latent generative models.

**Theoretical Claims:**

N/A

---

> ### Author Rebuttal · Authors · 2025-03-31
>
> We appreciate your insightful comments and efforts in reviewing our manuscript. Below, we provide our responses to each of your comments:
>
> ---
>
> **W1. EQ-VAE for T2I generation.**
>
> We appreciate the reviewer’s concern regarding the applicability of our method to text-to-image (T2I) generation. To address this, we conducted an additional T2I experiment using the MS-COCO dataset [1]. While this serves as a preliminary result in a small-scale setting—given that large-scale T2I experiments exceed our computational resources—it provides valuable empirical validation.
>
> For this experiment, we employed U-ViT-S/2 and followed their experimental setup [2]. We used SD-VAE to extract image latents in the baseline setting. During sampling, we use CFG with w=2.0. The table below reports gFID at every 50K iterations. We observe that EQ-VAE demonstrates improvements in T2I generation, highlighting the significance of equivariance regularization. These findings suggest that incorporating EQ-VAE into large-scale T2I models is a promising direction for future research.
>
>
> | **Iter.** | **U-ViT-S/2 w/ SD-VAE** | **U-ViT-S/2 w/ EQ-VAE** |
> |:--------:|:-------------------:|:-------------------:|
> | **50K**  |         15.5        |         12.4        |
> | **100K** |         8.6         |         7.6         |
> | **150K** |         7.7         |         7.1         |
> | **200K** |         7.5         |         6.9         |
> | **250K** |         7.3         |         6.8         |
> | **300K** |         7.2         |         6.7         |
> | **350K** |         7.1         |         6.6         |
> | **400K** |         7.0         |         6.6         |
> | **450K** |         7.0         |         6.5         |
>
> ---
>
>
> **W2.  Performance with EQ-VAE under same training resources as the baselines.**
>
> We appreciate the reviewer’s question about whether EQ-VAE solely accelerates convergence or also leads to better overall performance given the same training resources as the baselines. Due to our limited computational resources and short rebuttal timeline, we were unable to train XL models to full convergence (e.g. 1400 epochs for DiT/SiT). However, we recognize the importance of this evaluation and will include experiments with fully converged models in the final version of our paper to provide a more thorough assessment of convergence speed and final performance.
>
> That said, we emphasize that DiT-XL/2 with EQ-VAΕ, trained for only 300 epochs, already outperforms DiT-XL/2† (which uses SD-VAE before our EQ-VAE fine-tuning), even though the latter is considered fully converged after 1400 epochs (Table 4). This suggests that EQ-VAE is not only accelerating convergence but also contributing to improved performance. We will further investigate this in our updated experiments.
>
> ---
>
> **W3. Detailed Reconstruction Metrics.**
>
> We present detailed evaluation metrics for three models: SD-VAE, SD-VAE† (SD-VAE finetuned for 5 epochs with the original objective), and EQ-VAE (SD-VAE finetuned for 5 epochs with our objective). EQ-VAE significantly boosts the gFID performance of DiT-B/2. Importantly, this gain is achieved without sacrificing reconstruction quality as both EQ-VAE and SD-VAE† show similar improvements over the baseline SD-VAE across all the reconstruction metrics.
>
>
> | **Model**   | **gFID↓** | **rFID↓** | **PSNR↑** | **LPIPS↓** | **SSIM↑** |
> |:-----------:|:---------:|:---------:|:---------:|:----------:|:---------:|
> | **SDVAE**   |   43.5    |   0.90    |   25.82   |   0.146    |   0.71    |
> | **SD-VAE†** |   43.5    |   0.81    |   25.98   |   0.139    |   0.72    |
> | **EQ-VAE**  |   34.1    |   0.82    |   25.95   |   0.141    |   0.72    |
>
> ---
>
> **Q1. Autoencoder Training Steps.**
>
> We thank the reviewer for this question. We provide a detailed breakdown of the original training epochs for SD-VAE and SD-VAE-16, calculated based on the training steps reported in the [official repository](https://github.com/CompVis/latent-diffusion?tab=readme-ov-file#model-zoo) and the batch size specified in the [hugging face repository](https://huggingface.co/stabilityai/sd-vae-ft-ema-original). The epochs are calculated as follows epochs = (steps * batch_size) / dataset_size, where the dataset size for OpenImages is ~1.74M.
> For VQ-GAN, SD3-VAE, and SD-XL-VAE, to the best of our knowledge, their training iterations are not explicitly stated in their respective papers or official repositories.
>
>
> | **Model**       | **OpenImages Epochs** |
> |:-----------:|:-------------------:|
> | **SD-VAE**      | 27.1          |
> | **SD-VAE-16**   | 48.8           |
>
> We note that while our primary benchmark experiments use 5 fine-tuning epochs, our ablation study in Fig. 5 shows that even 1 epoch leads to notable performance improvements, particularly in terms of gFID.
>
> ---
>
> [1] Microsoft COCO: Common Objects in Context, In ECCV 2014
>
> [2] All are Worth Words: A ViT Backbone for Diffusion Models, In CVPR 2023

---

### Decision · Program_Chairs · 2025-05-01

**Decision:**

Accept (poster)

**Comment:**

This paper receives unanimously positive reviews for its solid contributions in introducing equivariance regularization into latent generative models. Although there are some relatively minor concerns around its evaluation and theoretical contributions, the work has made substantial contributions to generative modeling.

Therefore, I recommend acceptance of this work.